

# TopoCLIM: Rapid topography-based downscaling of regional climate model output in complex terrain v.1.0

Joel Fiddes[1], Kristoffer Aalstad[2], and Michael Lehning[1,3]

[1]WSL Institute for Snow and Avalanche Research SLF, Davos, Switzerland
[2]Department of Geosciences, University of Oslo, P.O. Box 1047, Blindern, 0316 Oslo, Norway
[3]CRYOS, School of Architecture, Civil and Environmental Engineering, Ecole Polytechnique Fédérale de Lausanne, Lausanne, Switzerland

**Correspondence:** Joel Fiddes (joel.fiddes@slf.ch)

**Abstract.** This study describes and evaluates a new downscaling scheme that specifically addresses the need for hillslope scale atmospheric forcing timeseries for modeling the local impact of regional climate change projections on the land surface in complex terrain. The method has a global scope in that it does not rely directly on surface observations and is able to generate the full suite of model forcing variables required for hydrological and land surface modeling at hourly timesteps. It achieves
this by utilising the previously published TopoSCALE scheme (Fiddes and Gruber, 2014) to generate synthetic observations of current climate at the hillslope scale, while accounting for a broad range of surface-atmosphere interactions. These synthetic observations are then used to debias (downscale) CORDEX climate variables using the quantile mapping method. A further temporal disaggregation step produces sub-daily fields. This approach has the advantages of other empirical-statistical methods, namely speed of use, while avoiding the need for ground data, which is often limited. It is therefore a suitable method for a
wide range of remote regions where ground data is absent, incomplete, or not of sufficient length. The approach is evaluated using a network of high elevation stations across the Swiss Alps and a test application of modelling climate change impacts on Alpine snow cover is given.

## 1   Introduction

Climate change has, and will continue to cause significant changes in the global cryosphere with increasing impacts likely in a
wide range of domains (Hock et al., 2019). Where observational records of sufficient length exist we are able to quantify these changes. Such records are often curated as part of national or international networks e.g. the World Meteorological Organisation's Global Cryosphere Watch, World Glacier Monitoring Service or the Global Climate Observing System. However, such locations are globally sparse, with particular observational gaps in remote regions due to technical difficulties and resources required to maintain monitoring infrastructure.

In order to obtain possible scenarios of future conditions we are reliant on climate models. However, for meaningful impact studies climate time-series are often required at higher spatial and temporal resolutions than currently available from Global or even Regional Climate Models (GCMs/ RCMs). This is especially the case in heterogeneous terrain such as mountain regions where topographic variability is high over short horizontal distances. Various methods of downscaling can be utilised





to achieve this goal. Dynamical downscaling typically applies an RCM or numerical weather prediction model (e.g. WRF) at

high resolution over a limited area in order to obtain more detailed process representation. This requires no additional data beyond a boundary forcing, yet is computationally costly (normally a supercomputer is required) and is therefore generally applied to limited domains and/or time periods. In addition, an extensive set of boundary fields are required in order to set up a run that are not normally available via standard distribution portals such as Earth System Grid Federation (ESGF), further complicating possible studies. Empirical-statistical downscaling (ESD) approaches are typically computationally cheap to run

yet require extensive and robust ground observations that are often either not available, of uncertain quality, or distributed unequally according to important gradients such as elevation. In addition, timeseries from many stations are not available at climate timescales - typically 30 years (Arguez and Vose, 2011), rendering the application of ESD methods problematic. Furthermore, many modern physically based impact models require a full suite of forcing variables to drive them, usually at sub-daily time-steps. Such requirements are rarely met by initiatives that have provided input for impact studies (Michel et al.,

2021). It is increasingly recognized that analysis of extremes, and not just mean values, is required to fully quantify the impact of climate change (Katz and Brown, 1992). By definition this requires highly temporally resolved forcing, as climatic extremes often occur over short timescales e.g daily maximum temperature or storm peak that require sub-daily simulations.

Climate model timeseries, even from the latest generation of RCMs, typically exhibit bias (systematic deviations) when evaluated against observations (Ivanov et al., 2018; Kotlarski et al., 2014). These biases need to be corrected before climate

timeseries can be used to force a locally applied impact model (Wood et al., 2004). However, in impact studies we are typically interested in a climate change signal (CCS) which is a quantitative measure of the difference between a future climate statistic and historical reference period (Themeßl et al., 2012). Bias correction (BC) can modify the CCS (Ivanov et al., 2018; Themeßl et al., 2012) which has been a subject of discussion and often seen as a deficiency in BC methods (Hempel et al., 2013). However, it is recognised that model biases typically do not cancel out in the calculation of a CCS and therefore its modification

under BC has been interpreted recently as an enhancement rather than a deficit, particularly in intensity dependent biases which characterise variables such as precipitation (Gobiet et al., 2015).

There are a wide range of BC methods (Gutmann et al., 2014) with perhaps the most established and widely used being quantile mapping (QM) which has been shown to perform favourably in comparison studies (Teutschbein and Seibert, 2013; Themeßl et al., 2011) and found to cope well with with non-stationary conditions - removing the restrictive stationarity as-

sumption in climate BC. It is also one of the few methods able to correct wet day frequency and intensity. QM is a distribution based BC method that removes quantile dependent biases with respect to a reference period (Ivanov and Kotlarski, 2017), it therefore corrects the variance and not just the mean. It should be noted that if the the climate variable and reference are at the same spatial scale a pure bias correction is applied, whereas if the reference is at a finer scale (e.g. a meteorological station) then an implicit downscaling is achieved, which also makes it a class of empirical-statistical downscaling (ESD) methods.

While concerns over the deterministic nature of QM (Maraun, 2013) and effect on multi-day statistics (Addor and Seibert, 2014) are acknowledged, it is widely accepted to be a pragmatic approach to satisfy the requirements of impact models (Rajczak et al., 2016) with deficits shared by all statistically-based methods. A useful overview of the issues involved with respect to impact modelling can be obtained from Stocker et al. (2015).





All ESD/BC methods require climate scale timeseries of observations (typically 30 years). In remote regions lacking suffi-

cient historical observations this requirement can be difficult to satisfy. Atmospheric reanalysis data-sets have been proposed as a means to compensate for missing or incomplete observations (Cao et al., 2019; Fiddes and Gruber, 2014) in order to provide a "best-guess" of the current state. Moreover, global reanalysis datasets can form the basis for impact studies with a global consistency.

In this study we address the problem of impact-model ready climate timeseries with a new modelling framework called

"TopoCLIM". We use the latest ECMWF global reanalysis dataset ERA5 together with the downscaling method TopoSCALE (Fiddes and Gruber, 2014) to provide a robust assessment of local-scale meteorological forcing for the reference period. Using these pseudo-observations we are able to debias climate timeseries in regions lacking ground observations. Furthermore, this method provides a full suite of forcing data required to run a numerical model at sub-daily timesteps.

By coupling this to the subgrid clustering scheme (Fiddes and Gruber, 2012) and the snow model FSM (Essery, 2015), we

demonstrate the ability of this scheme to efficiently generate high resolution (100 m) climate change maps of snow cover over the entire Swiss Alps. We test this scheme both with a detailed evaluation at the Weissfluhjoch meteorological station, and across the Swiss network of high elevation stations (IMIS) which has a large spatial coverage.

## 2  Methods

### 2.1  Overview

The scheme is implemented in Python with several specific sub-routines implemented in R (e.g. the quantile mapping package). An overview of the processing pipeline is given in Figure 1 and can be summarised as a three-step process: (1) A quasi-physical topography-based downscaling method TopoSCALE (Fiddes and Gruber, 2014) generates hillslope scale (defined by the DEM resolution) forcing time-series for the reference period from the ERA5 reanalysis, (2) the BC method quantile mapping (Gudmundsson et al., 2012) is used to statistically downscale (debias) a climate time series at the given point for

which we now have a reference from downscaled ERA5 forcing, (3) a disaggregation scheme (Förster et al., 2016) generates hourly climate timeseries based on observed sub-daily distributions of meteorological variables. We demonstrate this approach by downscaling CORDEX RCM data at both hillslope scale and additionally generalising this to a map product using the subgrid scheme, TopoSUB (Fiddes and Gruber, 2012). The overall philosophy of this approach is to develop methods that are global in application and therefore can be used in data-poor regions, are efficient to run and repeat (ability to be experimental),

and yet address the key drivers of climate-surface variability in complex terrain.

### 2.2  Preprocessing

The CORDEX data download is achieved using a custom tool built around the ESGF python client. This is not a trivial task due to the large data volumes involved and variable uptime of data nodes. All preprocessing steps of raw CORDEX data, such as concatenating NetCDF time series, extracting region of interest and regridding from rotated pole projections, was accomplished





using standard tools from the Climate Data Operators (CDO) suite. The CDO tools are incorporated into the preprocessing module of TopoCLIM and not used as standalone command line tools - enhancing the ease of use and reproducibility of the processing pipeline. As downloads are done by variable, it is possible that the full set of required forcing variables are not available for a given CORDEX model chain. If this is the case the model chain is excluded from any further processing.

Climate model calendars are often simplified for numerical reasons and are inherited from the parent GCM, these need to be correctly handled to produce comparable timeseries. Three calendars exist in the CORDEX data used here: "360-day" (every month is 30 days long), "365-day" (no leap year), and "standard" (complete Gregorian calendar). We convert all calendars to "standard" by linearly scaling dates to the standard Gregorian calendar and then gap filling missing data by linear interpolation. For example conversion from a "360-day" to a "standard" calendar, the output from the linear scaling will result in a 365 day timeseries (in the case of non-leap year) and be missing the following dates: January 31st, March 31st, June 1st, July 31st, September 31st and November 30th. In a second step these dates are gap-filled using linear interpolation.

### 2.3 Spatial downscaling of observations

ERA5 reanalysis (Hersbach et al., 2020) "observations" are downscaled using the TopoSCALE scheme (Fiddes and Gruber, 2014) to be used as the reference data in the bias correction. We acknowledge that reanalyses are not true observations (Parker, 2016), yet by assimilating an extensive set of observations into an NWP model, reanalyses are often considered to give the best possible view of the global climate (Dee et al., 2014). TopoSCALE performs a 3D interpolation of atmospheric fields available on pressure levels, to account for time varying lapse rates, and a topographic correction of radiative fluxes. The latter includes a cosine correction of incident direct shortwave radiation on a slope, adjustment of diffuse shortwave and longwave radiation by the sky view factor, and elevation correction of both longwave and direct shortwave. It has been extensively tested in various geographical regions and applications e.g. permafrost in the European Alps (Fiddes et al., 2015), permafrost in the North Atlantic region (Westermann et al., 2015), Northern hemisphere permafrost (Obu et al., 2019), Antarctic permafrost (Obu et al., 2020), Arctic snow cover (Aalstad et al., 2018), Arctic climate change (Schuler and Østby, 2020), and Alpine snow cover (Fiddes et al., 2019). This approach enables us to provide a climate length pseudo-observation timeseries globally, while accounting for the main topographic effects on atmospheric forcing. We call this product T-MET throughout the text. It should be noted that this serves as our reference throughout the paper. A detailed validation and quantification of uncertainty of the TopoSCALE method is given in Fiddes and Gruber (2014) and therefore is not repeated here. We do however compare our results to the station data variables air temperature and snow depth across the IMIS network.

### 2.4 Quantile mapping

Bias correction through quantile mapping $Q : x \rightarrow x^{\star}$ is achieved as follows (Panofsky and Brier, 1968)

$$x^{\star} = Q(x) = F_o^{-1} \left[ F_m \left( x \right) \right], \tag{1}$$

where the debiased output variable $x^{\star}$ is obtained by applying the quantile mapping function $Q$ to the biased input variable $x$. This function is generally formed through the composition of the modeled cumulative distribution function ($F_m$) and the





inverse of the observed cumulative distribution function ($F_o$). In our case, $F_o$ is obtained from the pseudo-observations in the form of downscaled ERA5 data, while $F_m$ is generated from the CORDEX output which we want to bias correct.

We use the R package QMAP for this purpose (Gudmundsson et al., 2012). Gudmundsson et al. (2012) compared different
implementations of QM for daily precipitation data and found that a non-parametric empirical approach (as implemented in the cited package) outperforms implementations relying on theoretical distributional assumptions. While quantile mapping ensures that quantile biases are corrected in the CDF it does not account for seasonally varying bias. It is therefore well suited to air temperature where we can be reasonably sure that winters are cold and summers hot, at least in mid to high latitudes. However, with precipitation the intra-annual distribution can be biased while the CDF may look reasonable (e.g. wet and
dry season timing could be shifted). We address this with a two step approach called QMAP_MONTH. We split the data according to 12 temporal subsets corresponding to the months of the calendar year and run the QM algorithm on each subset, computing QM parameters separately for each month. These are applied throughout the historical and climate time series at the appropriate months. Similar approaches have been used successfully in other studies e.g. Hanzer et al. (2018). Quantile mapping is performed on a subset of data during 1980-1995 to allow an evaluation to be performed over the time-period
135   1996-2006.

## 2.5   Temporal downscaling of observations

The final step in preparing the climate forcing is a temporal disaggregation to generate required sub-daily fields. Original hourly resolution T-MET are used to temporally disaggregate the downscaled (quantile mapped) daily climate timeseries. An adapted version of the Melodist package is used for this purpose (Förster et al., 2016). This disaggregates daily data based on observed
sub-daily distributions. It should be noted that this assumes sub-daily distributions are stationary in a future climate, which quite possibly may not be the case. However, a greater source of uncertainty likely exists in the ability of ERA5 to reproduce short timescale local weather patterns which require convection resolving model resolutions (Liu et al., 2017) and appropriate physics.

## 2.6   Simulation and maps

We use the Factorial Snow Model (FSM) model (Essery, 2015) to simulate the snow cover and TopoSUB (Fiddes and Gruber, 2012) to spatialise results to a 2D map. Briefly, TopoSUB is a topographic sampling scheme that reduces distributed modelling problems that explicitly model individual pixels to a lumped model several orders of magnitude smaller while considering the full range of topographic heterogeneity that exists. It achieves this by using a $k$-means clustering algorithm to perform a multidimensional classification on the model domain resulting in so-called TopoSUB clusters, consisting of pixels with similar
terrain parameters. In this way it is possible to produce high resolution maps (e.g. DEM resolution) over large (regional) modelling domains, while explicitly including important drivers of surface-atmosphere processes. The scheme is implemented on an HPC cluster for efficiency in an "embarrassingly parallel" sense i.e. no communication required between compute nodes. Typical setups use 100 nodes with run-times measured in minutes for full 1980-2100 runs. The scheme also has a desktop mode which typically utilises 8 cores and typical runs will require a few hours. These indicative numbers are provided merely





to give the reader an order-of-magnitude idea of how frugal this scheme is in terms of computation resources compared to dynamical downscaling.

## 2.7 Treatment of glaciers

Glacier zones are typically masked in studies of seasonal snow unless a glacier layer is explicitly accounted for in the model. In this study, however, we wanted to highlight and track changes in the glacier accumulation zones. These zones are areas where

the annual surface mass balance is positive, leading to the formation or growth of glaciers. With our framework we are able to identify such zones, as we are able to model areas with perennial snow cover with FSM and these give a good indication of glacier accumulation zones under a given climate. However, we ignore glacier dynamics so we are not able to adequately map the spatio-temporal evolution of glaciers.

## 3 Study region and data

### 3.1 Model domain

We consider two scales in this study (a) point-scale (meteorological stations: Weissfluhjoch and the IMIS network) and (b) regional (Swiss Alps), in order to illustrate typical applications of the scheme. A map of the study region and the location of the stations we used is given in Figure 2.

### 3.2 Climate data

The basic forcing comes from the regional climate model project CORDEX EUR-44 product (Jacob et al., 2014) at a nominal resolution of 44km. The 44km product was chosen over the 22km product as this had many more model chains available. The EURO domain also has an 11km product but this is not available globally, and therefore is not fit for the purpose of this study. Data was retrieved from the ESGF using an API and automated python based tool developed in this study. This is an important step as the number of file downloads are large with dimensions being models $\times$ variables $\times$ scenarios $\times$ time periods. The fact

that the ESGF consists of a distributed set of data nodes with variable uptime, further complicates the download process. We use daily data to force the scheme and retrieve historical data plus projections from two climate change scenarios, RCP2.6 ( a "very stringent" pathway, RCP2.6 requires that $CO_2$ emissions start declining by 2020 and go to zero by 2100) and RCP8.5 (emissions continue to rise throughout the 21st century). A full description of CORDEX datasets and models used is given in Table 2. Daily data was chosen for the method (therefore requiring a temporal disaggregation step) as limited number of model

chains are available at sub-daily resolutions. This is particularly the case outside of the EURO domain where use-cases for this method are envisaged. A higher number of model chains increases confidence in our results, by improved quantification of inter-model variability.



## 3.3 Reanalysis data

We use ECWMF's latest reanalysis product ERA5 (Hersbach et al., 2020), which uses version 41r2 of IFS which is the ECMWF
NWP model. ERA5 represents an evolution over its predecessor, ERA-Interim, by increasing the model spatial resolution to
30km, temporal resolution to hourly and the vertical model levels to 137. These reanalysis data are downscaled for the purpose
of bias correction using the TopoSCALE scheme (Fiddes and Gruber, 2014) as described above (cf. Methods).

## 3.4 Topography

NASA's SRTM-3 90m digital elevation model (DEM) is used as a topographical surface for TopoSCALE downscaling routines.
Slope, aspect and sky-view factor (the portion of the sky hemisphere that is visible for a given DEM pixel) are derived (Dozier
and Frew, 1990). A higher resolution DEM may also be used but likely does not add value as processes such as wind transport
that operate on these scales are not included in the model (Mott et al., 2018). Importantly, in our scheme, higher resolution does
not necessarily increase runtime either in point mode (trivially) or in spatial mode where the run-time in the main programme
modules is related to number of TopoSUB clusters. Additionally, the scheme is designed to scale well by generating cluster
forcings through array computations.

## 4   Simulation setup and evaluation

Predictive methods must by definition be evaluated on independent data from that which was used for calibration in order to
correctly evaluate how applicable a model is beyond the data-space within which it was developed. Further, highly adaptable
methods, such as the non-parametric techniques used in this study, are prone to over fitting. These issues are avoided in this
study as we perform an independent evaluation using station data from the IMIS station network, whereas calibration or
in this case downscaling, is performed using downscaled ERA5 fields. Note, none of the meteorological fields of the IMIS
stations were assimilated during the production of ERA5. Snow depth results are evaluated by automatic snow depth (cm)
measurements performed by sonic ranger (Campbell Scientific SR50), available from the Inter-cantonal Measurement and
Information System (IMIS) station network at 30 minute intervals. This is a high elevation station network that forms the
backbone of the national avalanche service in Switzerland.

The analysis in this study is organised as follows: The reference period is defined as 1981-2010 and future scenarios are
analysed for climate periods 2031-60 and 2070-100. We assess the downscaling of all meteorological fields at the Weissfluhjoch
IMIS station (Figure 3) which has the full suite of variables produced as compared to standard IMIS stations which lack a full
radiation balance. Here, we additionally test the performance of the scheme at point-scale (Figure 5 and 6). We further assess
air temperature and snow depth (as a proxy for precipitation) using the entire IMIS network (Figure 6). We do not assess
precipitation directly as year-round datasets are not available from the IMIS network due to unheated gauges, and snow depth
is often considered a more robust variable to measure at high elevation. After this we provide results using the scheme at
various spatial scales (Figure 6-8).





## 5    Results and discussion

### 5.1    Station evaluation of point forcing timeseries

Figure 3 and Table 3 and 4 (statistics) shows an evaluation of the quantile mapping scheme at the WFJ station both as a cumulative distribution function (left column) and a day of year (DOY) plot which averages all values in the timeseries for a given DOY (right column). Here we compare grid-box CORDEX ensemble mean, a single parameter set quantile map run (QM_QM) and monthly parameter set quantile map run (QM_MONTH) to T-MET and station measurements. This comparison

is done over the period 1996-2006 which corresponds to the overlapping time-frames of CORDEX-HIST, T-MET and station data, and also importantly does not include the period over which quantile mapping is run. We perform two sets of comparison, first with T-MET as a reference as this is the target of the quantile mapping (Table 3). This shows how the scheme behaves particularly in terms of the different QM and QM_MONTH implementations (Table 3). In the second, we compare results directly to the measurements at the station to get a global look at how the scheme performs (Table 4), of course this is then

subject to residual erro in the TopoSCALE downscaling scheme which produces T-CLIM (Table 5) and therefore must be interpreted carefully.

With respect to target T-CLIM (Figure 3, Table 3) Percentage bias and RMSE are generally strongly decreased by the scheme in the standard QMAP mode and further by the QMAP_MONTH variant, especially where there is a time varying bias signal (e.g. shortwave radiation).

With respect to station measurements (Table 4) we see overall improvement in statistics by the scheme, however this is not always reflected in an improvement in statistics between QM and QM_MONTH as there is residual error between the station measurements and downscaled T-MET. The strongest improvements are variables which are downscaled according to model pressure levels in the TopoSCALE scheme (air temperature and relative humidity). Precipitation is the only parameter we do not see an improvement in the scheme with respect to the measurements, but this is expected due to high uncertainty in

precipitation and the fact this is not addressed by the base TopoSCALE downscaling. This point is further discussed below in Section 5.4.

Figure 4 shows a typical point scale application generating a forcing timeseries, in this case for air temperature. The effect of the bias correction is shown with a clear improvement with respect to T-MET. Figure 5 gives a point scale example of snow height evolution at WFJ. Available snow height observations from WFJ show good agreement with both the historical period

and first decade of the RCP runs. Note the stable/rising snow height under RCP2.6 by end of century correlated to stabilised mid-century temperatures shown in Figure 4.

### 5.2    Spatial evaluation of snow depth

Figure 6 gives the evolution of snow depth averaged across the IMIS network in terms of mean DOY snow depth, starting 1 September. IMIS station measurements are given as reference, however it should be noted that the time-period covered by

each station is variable. The shortest station record is 10 years, therefore the dataset nominally represents the period 1996-2018. These are compared to T-MET snow height by only using days present in the IMIS dataset. At this synoptic spatial and





temporal scale there is good agreement with low bias and RMSE scores and high correlation (Table 4). The evolution of snow height for future climate scenarios and two future periods is given. By mid-century under RCP2.6 a reduction in peak snow depth of around 20cm is seen and further 25cm under RCP8.5 with respect to HIST. Peak snow depth occurs around 30-40 days

earlier. By late century RCP2.6 snow depth has not further deteriorated and in fact shown signs of possible recovery with peak snow depth moving towards that of the HIST period. RCP8.5, however, gives a strong further reduction in peak snow depth of around 1 m with respect to HIST. The comparison between HIST (1980-2006) and T-MET/ IMIS (1996-2018) is useful but should be treated with caution due to only partially overlapping periods (defined by availability of station measurements). HIST has a higher peak snow depth and later snow melt-out date, which could be due to less climate change as it represents an

earlier period than IMIS and T-MET.

### 5.3   TopoCLIM application: Climate change impacts on Alpine snow cover

The results in Figure 7 were generated by coupling TopoCLIM with the TopoSUB spatial framework to generate transiently modelled snow height maps at 100m resolution. The ensemble mean is used in each scenario/ time-period plot. We highlight again that by using this simple approach we implicitly model climate-viable glacier accumulation zones, where the snowpack

does not melt-out by the end of summer. Mid-century results are comparable between RCP2.6 and RCP8.5 with marginal snow cover affected in RCP8.5 (except for the lower elevation Jura mountains) and decrease in accumulation zones. By late-century the difference is strong as also seen in Figure 6 with a strong increase in the snowline elevation and reduction of snow height at high elevations. Glacier accumulation zones remain only in the high Valais (Mattertal) and Bernese Oberland around today's accumulation zone of the Aletsch Glacier. Figure 8 provides a more quantitative picture of snow cover-elevation dynamics.

This hypsometry plot summarizes the pixels of Figure 7 by showing the mean and standard deviation of snow depth at each 50 m elevation band across the entire domain for three time periods and scenarios RCP2.6/8.5. As expected, a strong decrease in snow depth is seen at all elevations under RCP8.5. A slightly different story is seen under RCP2.6 with snow depth reducing up to mid-century followed by an increase at high elevations (above 3500masl) by end of the century - this is a consistent message throughout this study and reflects the results of Figures 5, 6 and 7. It can be partially explained by stabilisation of air

temperatures by mid-century (Figure 4) together with increased precipitation in the Alpine region (Jacob et al., 2014; Smiatek et al., 2016). An interesting observation in this figure is that in all time periods and scenarios snow depth is limited both at low elevation by temperature and at high elevation by terrain, which tends to be steeper and therefore permits lower accumulations due to avalanching (the latter effect is explicitly accounted for in the modelling scheme by removing snow linearly above a slope threshold (c.f. Fiddes et al., 2015). A final point of note in this figure is the truncation level indicating the accumulation

zone elevation which is approximately 4000 m for HIST, above this level seasonal snow is not possible, as it will not melt before the following winter season or the ground is too steep for snow accumulation. This upper limit of seasonal snow limit rises to around 4500masl over the 21st century, well above former glaciated surfaces. An implication of this for water resources is that while we lose a large quantity of glacier accumulation zones during the 21st Century, irrespective of scenario, we will likely not lose seasonal snow water resources at those elevations.



## 5.4 Forcing uncertainty

The largest source of uncertainty in the scheme is the reanalysis forcing from ERA5. Both quantile mapping and disaggregation of fields to sub-daily timesteps are inherently constrained by the distribution of ERA5 fields. While ERA5 provides hourly data and therefore resolves the diurnal cycle, it remains a 25 km model with a correspondingly smooth topographical surface representation and parameterisation of physical processes that occur on shorter length scales. Typical examples are convective precipitation and cold air pooling in valley bottoms (Cao et al., 2017; Liu et al., 2017), orographic enhancement of precipitation and wind fields (Gerber et al., 2018; Mott et al., 2018; Gutmann et al., 2016). As the density of observations that are assimilated in reanalyses varies globally we expect the performance of the TopoCLIM model pipeline to be a function of how well constrained ERA5 is in any given location. A full analysis and discussion of TopoSCALE uncertainties is given by Fiddes and Gruber (2014) and to some extent in Fiddes et al. (2019). The most uncertain variable is precipitation which is clearly a critical point for snow modelling studies. We do however show that there are no large scale biases in the precipitation field at least in our snow height comparison across the IMIS network. We have shown in previous studies that variables driving the energy balance (TA, ILWR, ISWR) are downscaled with good skill by TopoSCALE. One method we have explored to reduce (and quantify) meteorological forcing uncertainty and precipitation uncertainty in particular, is through Bayesian data assimilation of globally available satellite products using a particle batch smoother, which has shown promising results (Fiddes et al., 2019; Alonso-González et al., 2020). The coupling of this scheme with TopoCLIM will be the subject of subsequent work.

## 5.5 Evaluation uncertainty

Our station measurements are characterised by considerable uncertainties, a problem that is particularly acute for precipitation related fields such as snow depth, so we have to some degree a chicken and egg scenario when it comes to model validation, in that it's hard to untangle the origin of apparent errors. For example, stations tend to be situated in sheltered flat to concave topography. Here we expect there to be considerable preferential deposition from surrounding windblown slopes and ridgelines (Grünewald and Lehning, 2015). In general we would expect stations therefore to be positively biased with respect to large scale precipitation fluxes. Additionally, certain stations will be exposed to very local climatic effects which are not represented in the large scale 25 km resolution ERA5 forcing. Examples of such local effects include wind funnelling leading to scouring, enhanced Foehn effects and local orographic enhancement.

## 5.6 Snow model uncertainty

The snow model used in this study, FSM, is an intermediate complexity physically-based model and we do not expect it to perform as well with respect to snow densification as a more complex snow physics model such as SNOWPACK (Lehning et al., 2002; Wever et al., 2015), this can introduce uncertainty when using snow depth as a validation parameter. However, it should be noted that there is active discussion about snow model complexity and how this does not necessarily lead to improved performance (c.f. Magnusson et al., 2015). Snow water equivalent is a simpler modelling objective but a much harder





measurement objective and therefore few sites are available - particularly with good coverage at regional scales, therefore limiting its applicability for large-scale evaluations.

## 6    Conclusions

In this study we have developed and tested a new scheme for downscaling regional climate projections, specifically designed
to provide hillslope scale forcings for impact models. We take advantage of the now globally available CORDEX RCM data to develop a method with global scope. The scheme is parsimonious and adheres to the philosophy of TopoSCALE upon which it builds, that is modelling tools that bridge the gap between relatively simple empirical approaches and full dynamical models that require extensive computing resources. It can be run both on desktop or cluster environments. The target application of this scheme is impact modelling in complex terrain where significant atmosphere-surface interactions need to be considered.
A particular application is in remote areas where ground data may either not be present at all or not available for the duration of a climate normal period, meaning that traditional ESD methods are problematic to use. Another strength of this approach is that it produces continuous timeseries, such that it permits transient simulations in contrast to other parsimonious methods such as the delta-change approach, an important point for domains such as soil, ground-ice or glaciers where surface forcings drive processes over decadal timescales.
This framework is adaptable to any kind of meteorological input data (both the reference data and the future/past period). Here we have given an example with the reference based on downscaling ERA5, but it could equally be generated by downscaling other reanalyses such as MERRA or outputs from regional models such as WRF, COSMO, or ICAR. The method would also be applicable to other future projections such as those from CMIP6 which are coming online now. Exploring all these possibilities is of course beyond the scope of one paper, but it's important to emphasize the modularity of this framework.
Another aspect to highlight is that we are downscaling an ensemble of CORDEX outputs which also gives us a better idea of the uncertainty in future climate projections and impacts on the snowpack (in this case). It should also be stated that as a framework other downscaling routines or bias correction approaches could be used, our main message is that the combination of these methods provides a powerful way of rapidly obtaining hillslope scale climate forcings anywhere on the globe.

As a final point of note, the scheme could also be usefully applied retrospectively to bias correct historical reanalysis that
suffer from uncertainties related to pre-satellite era data assimilation and a much reduced surface station network, such as ECMWFs 20th century reanalysis, ERA-20CM (1900-2010).

*Code and data availability.*   Model code, documentation and example used in this study is archived at 10.16904/envidat.229 and Github repository is available at: https://github.com/joelfiddes/topoCLIM



*Author contributions.* JF devised the study conducted the analysis and wrote the manuscript, KA assisted with algorithm development and
text, ML helped devise the study, assisted with analysis and text.

*Competing interests.* We declare no competing interests.

*Acknowledgements.* This study was conducted within the SNF project "Precipitation in extreme environments" and the WSL programme Climate Change impacts on Alpine Mass Movements (CCAMM). KA was funded by the European Space Agency Permafrost$_C CI project (https : //climate.esa.int/en/projects/permafrost/), and acknowledges support from the LATICE strategic research area at the University of Oslo.$
We acknowledge the World Climate Research Programme's Working Group on Regional Climate, and the Working Group
on Coupled Modelling, former coordinating body of CORDEX and responsible panel for CMIP5. We also thank the climate
modelling groups (listed in Table 2 of this paper) for producing and making available their model output. We also acknowledge
the Earth System Grid Federation infrastructure an international effort led by the U.S. Department of Energy's Program for
Climate Model Diagnosis and Intercomparison, the European Network for Earth System Modelling and other partners in the
Global Organisation for Earth System Science Portals (GO-ESSP).





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



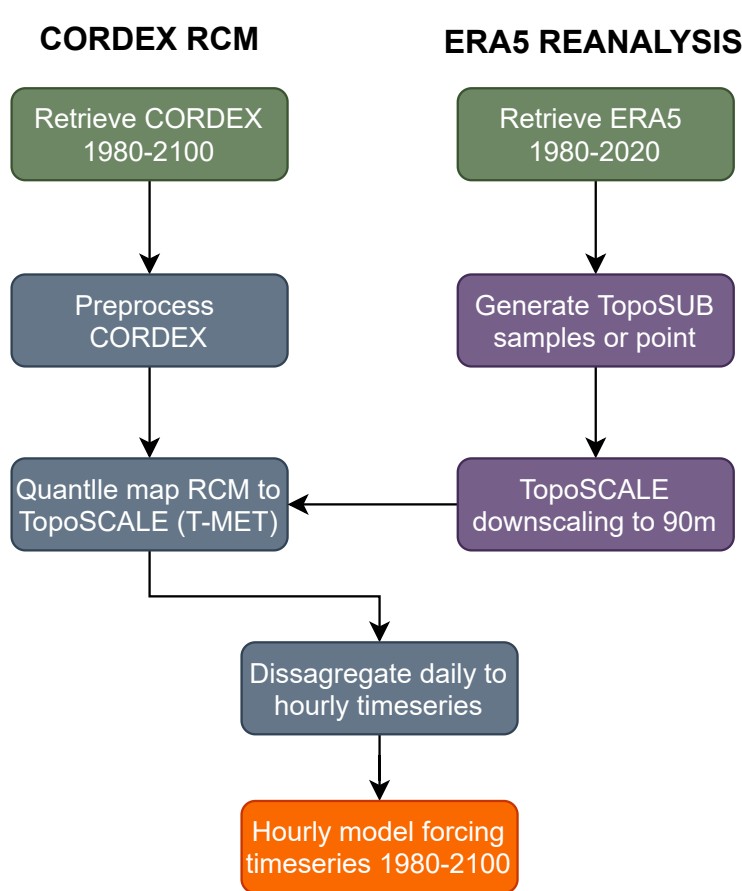

**Figure 1.** Schematic overview of the main TopoCLIM processes.



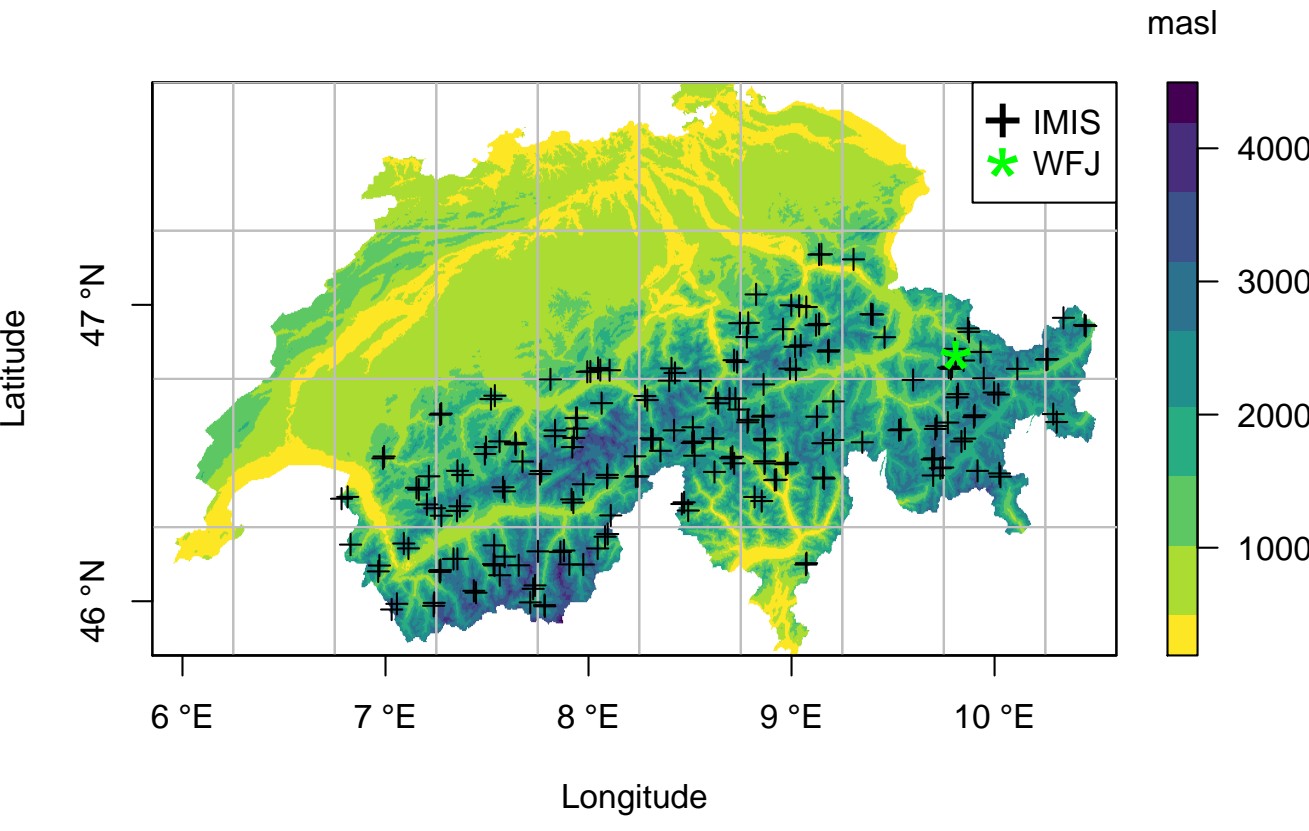

**Figure 2.** Study domain map with CORDEX grid overlaid. IMIS stations and the Weissfluhjoch site (WFJ) are indicated.



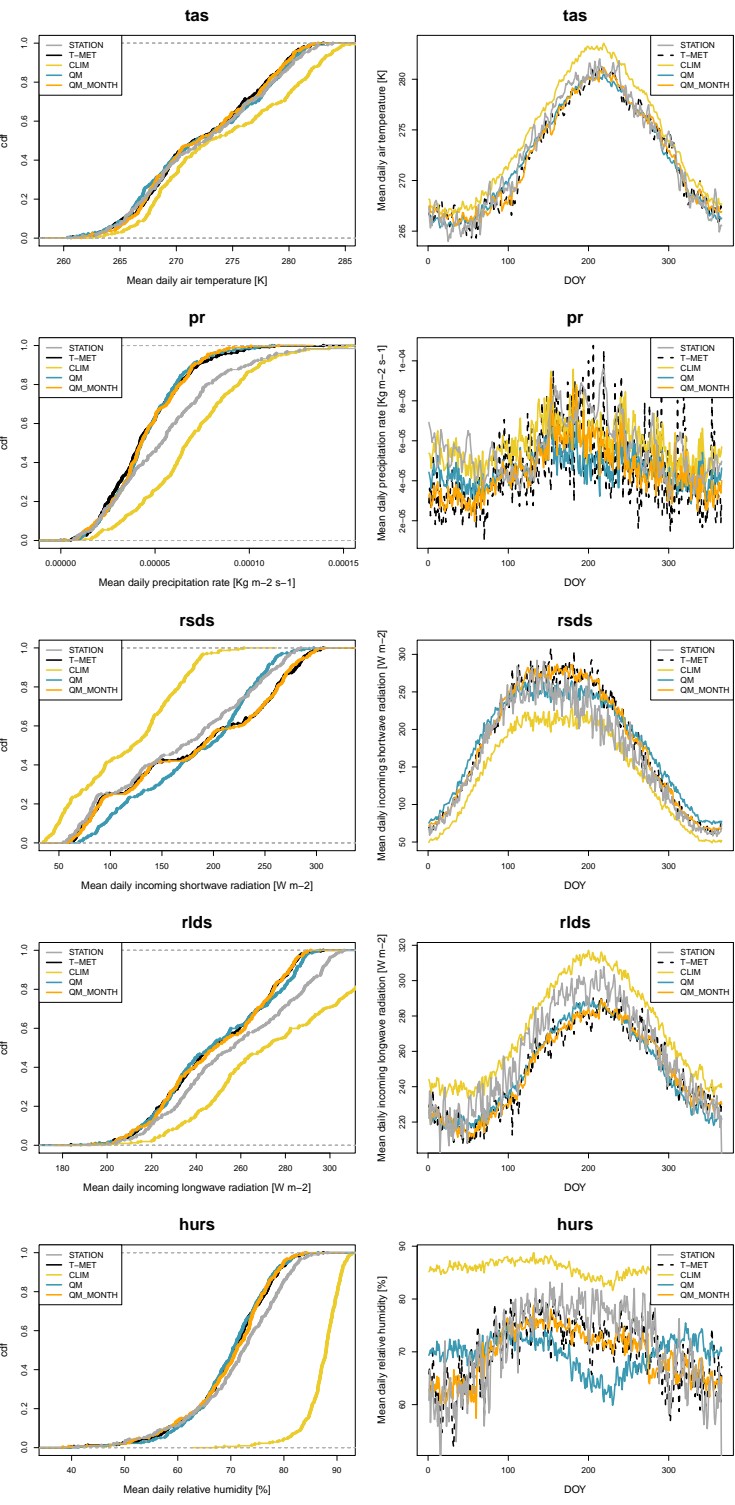

**Figure 3.** Evaluation of the quantile mapping routine at the Weissfluhjoch station in standard mode "QM_QM" (single parameter set) and "QM_MONTH" seasonally varying parameter set. CLIM is the uncorrected CORDEX data. STATION is the Weissfluhjoch station measurements. T-MET is the downscaled ERA5 data obtained using TopoSCALE. Shown are the cumulative density function over the period 1981-2010 (left panel) and seasonal distribution given as average by day of year (DOY) for the same period (right panel).



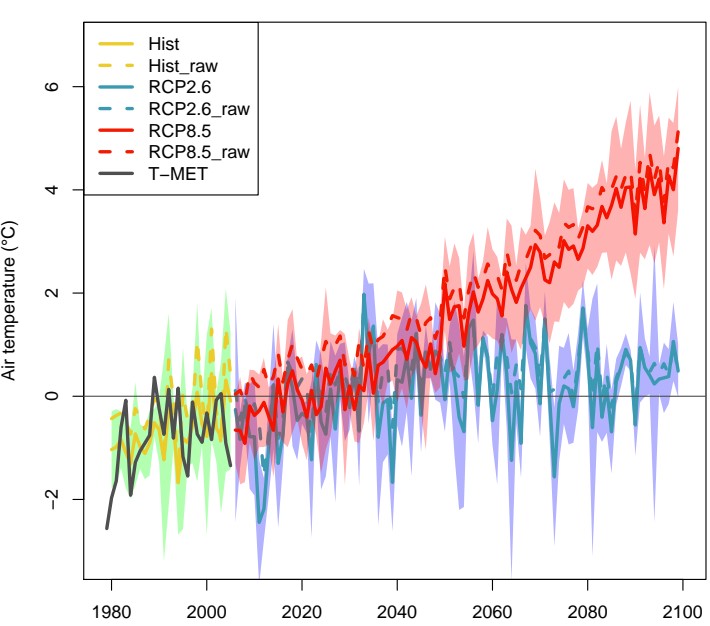

**Figure 4.** A point-scale TopoCLIM product: Mean annual near surface air temperature at the Weissfluhjoch (2540 masl) showing corrected historical, RCP2.6 and RCP8.5 time series. T-MET and uncorrected CORDEX data are also shown for comparison. The coloured envelopes indicates the model spread and ensemble mean is given by the bold line. The zero degree isotherm is given by the horizontal line.



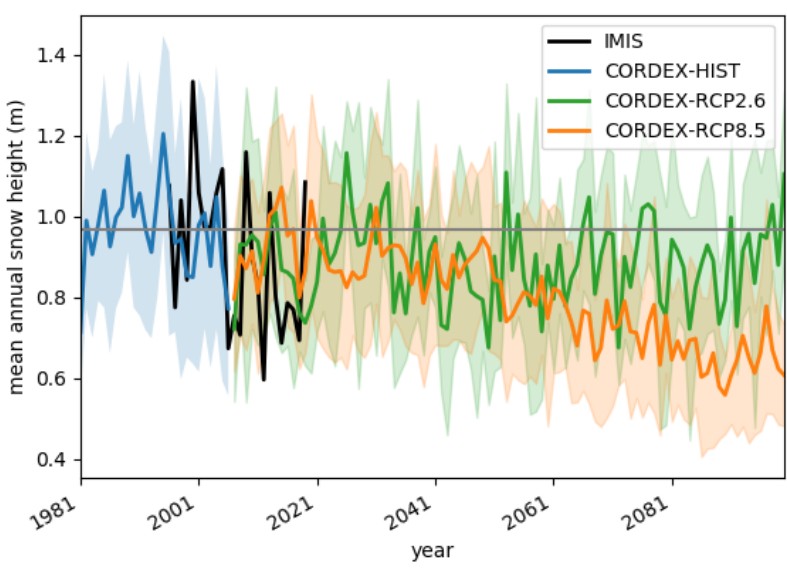

**Figure 5.** An example point-scale TopoCLIM product: Mean annual snow depth (m) ensemble at the Weissfluhjoch (2540m asl) for historical and future RCP scenarios. Observations from WFJ2 are given in black for a qualitative comparison. Ensemble means are indicated by solid lines and long term average over the plotted historical period (1981-2010) is given by the horizontal line. Observations lie within the ensemble spread, indicating that the bias correction method works satisfactorily. It should be stressed that we do not attempt a quantitative comparison here as CORDEX (and climate models more generally) variability is not expected to be perfectly synchronised with observed variability. Nonetheless, we still prefer to present the inter-annual variability that is present in both the model and the observations instead of showing decadal means.



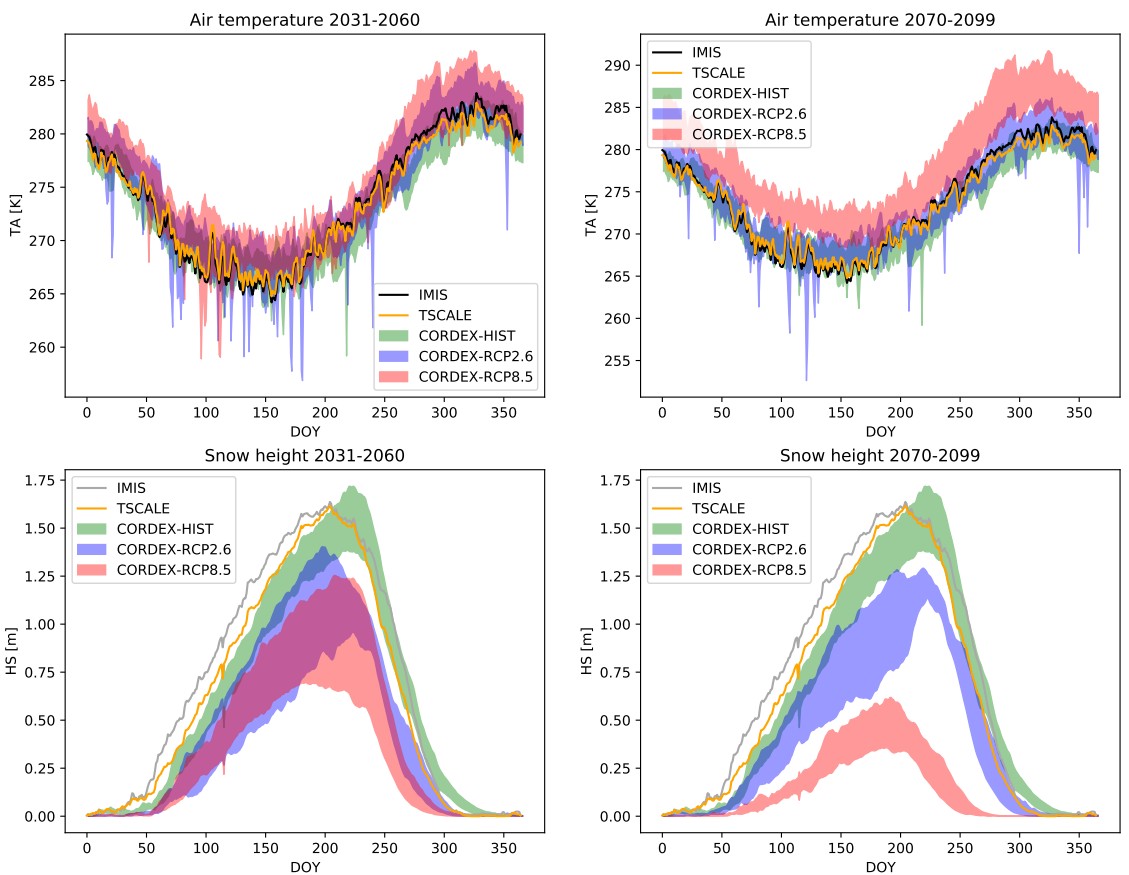

**Figure 6.** Large scale evaluation of the scheme: Mean DOY air temperature and snow height across all IMIS stations (IMIS) is compared to the same data generated by the TopoSCALE downscaling of ERA5 (TSCALE) and TopoCLIM downscaled CORDEX data for the historical period (1981-2010) and scenarios RCP2.6/8.5 for near (2031-2060) and far future periods (2070-2099). The full width of the model ensemble is represented by the shading. Note TSCALE and IMIS have the same time-frame (variable by station between 1996-2020) and are therefore directly comparable (see Table 5), but this only partially overlaps with the CORDEX historical period and therefore is intended for visual comparison only.



**Figure 7.** Example TopoCLIM climate change maps: Mean snow depth (m) for RCP2.6 and RCP8.5 (columns) and time periods 1981-2010, 2031-2060 and 2070-2099 (rows). Perennial snow-pack (does not melt in an annual cycle) is masked out (white regions) and assumed to represent climatological glacier accumulation zones.

**Figure 8.** Hypsometry of snow depth over the Swiss Alps for RCP2.6, RCP8.5 for periods 1981-2010, 2031-60 and 2070-99. This data is derived from the ensemble mean, with shading showing +/- 1 standard deviation of values in each elevation band. The elevation limit in each time-period corresponds to the threshold between seasonal and non-seasonal snow, or glacier accumulation zones. This demonstrates the rising equilibrium-line altitude of glaciers over the 21st century in all RCP's and extension of seasonal snow into the former glacier zones.





**Table 1.** CORDEX variables used in this study, together with the Climate and Forecast Conventions (CF) standard name .

| Variable name | Units | Timestep [h] | CF ong name | CF standard name |
|---|---|---|---|---|
| tas | K | 3 | Near-Surface Air Temperature | air_temperature |
| pr | kg m$^{-2}$ s$^{-1}$ | 3 | Precipitation | precipitation_flux |
| ps | Pa | 3 | Surface Air Pressure | surface_air_pressure |
| hurs | % | 3 | Near-Surface Relative | Humidity relative_humidity |
| rsds | W m$^{-2}$ | 3 | Surface Downwelling Shortwave Radiation | surface_downwelling_shortwave_flux_in_air |
| rlds | W m$^{-2}$ | 3 | Surface Downwelling Longwave Radiation | surface_downwelling_longwave_flux_in_air |
| uas | m s$^{-1}$ | 6 | Eastward Near-Surface Wind | eastward_wind |
| vas | m s$^{-1}$ | 6 | Northward Near-Surface Wind | northward_wind |

**Table 2.** CORDEX model chains used in this study.

| GCM | RCM | Scenario | Ensemble | Version |
|---|---|---|---|---|
| CNRM-CERFACS-CNRM-CM5 | CLMcom-CCLM5-0-6 | Hist/RCP2.6/8.5 | r1i1p1 | v1 |
| CNRM-CERFACS-CNRM-CM5 | SMHI-RCA4 | Hist/RCP2.6/8.5 | r1i1p1 | v1 |
| ICHEC-EC-EARTH | CLMcom-CCLM5-0-6 | Hist/RCP2.6/8.5 | r12i1p1 | v1 |
| ICHEC-EC-EARTH | KNMI-RACMO22E | Hist/RCP2.6/8.5 | r12i1p1 | v1 |
| ICHEC-EC-EARTH | SMHI-RCA4 | Hist/RCP2.6/8.5 | r12i1p1 | v1 |
| MIROC-MIROC5 | CLMcom-CCLM5-0-6 | Hist/RCP2.6/8.5 | r12i1p1 | v1 |
| MPI-M-MPI-ESM-LR | SMHI-RCA4 | Hist/RCP2.6/8.5 | r12i1p1 | v1 |
| NCC-NorESM1-M | SMHI-RCA4 | Hist/RCP2.6/8.5 | r1i1p1 | v1 |
| NOAA-GFDL-GFDL-ESM2M | SMHI-RCA4 | Hist/RCP2.6/8.5 | r1i1p1 | v1 |





**Table 3.** Statistics from the evaluation in Figure 3. Correlation (R), root mean squared error (RMSE) and percentage bias (PBIAS) are given relative to downscaled ERA5 data T-CLIM.

| Statistic | Scheme | tas [K] | pr [kg m$^{-2}$ s$^{-1}$] | rsds [W m$^{-2}$] | rlds [W m$^{-2}$] | hurs [%] |
|-----------|--------|---------|---------------------------|-------------------|-------------------|----------|
| R [-] | CLIM | 0.98 | 0.39 | 0.99 | 0.95 | 0.02 |
| R [-] | QM | 0.98 | 0.35 | 0.98 | 0.95 | -0.11 |
| R [-] | QM_MONTH | 0.99 | 0.55 | 0.99 | 0.97 | 0.74 |
| RMSE | CLIM | 2.27 | 2.17e-05 | 43.00 | 24.19 | 17.29 |
| RMSE | QM | 1.04 | 1.75e-05 | 19.09 | 7.41 | 6.89 |
| RMSE | QM_MONTH | 0.84 | 1.58e-05 | 10.46 | 5.67 | 3.94 |
| PBIAS [%] | CLIM | 0.7 | 28.8 | -21.1 | 9.1 | 23.3 |
| PBIAS [%] | QM | 0 | 0.3 | 0.1 | 0.1 | -0.1 |
| PBIAS [%] | QM_MONTH | 0 | 0.2 | 0 | 0.1 | -0.1 |

**Table 4.** Statistics from the evaluation in Figure 4. Correlation (R), root mean squared error (RMSE) and percentage bias (PBIAS) are given relative to station measurements at Weissfluhjoch (STATION).

| Statistic | Scheme | tas [K] | pr [kg m$^{-2}$ s$^{-1}$] | rsds [W m$^{-2}$] | rlds [W m$^{-2}$] | hurs [%] |
|-----------|--------|---------|---------------------------|-------------------|-------------------|----------|
| R [-] | CLIM | 0.99 | 0.22 | 0.97 | 0.97 | -0.16 |
| R [-] | QM | 0.98 | 0.20 | 0.97 | 0.96 | -0.30 |
| R [-] | QM_MONTH | 0.98 | 0.32 | 0.97 | 0.94 | 0.76 |
| R [-] | T-CLIM | 0.98 | 0.35 | 0.97 | 0.94 | 0.76 |
| RMSE | CLIM | 1.86 | 2.59e-05 | 29.41 | 15.61 | 16.94 |
| RMSE | QM | 1.01 | 2.82e-05 | 22.89 | 11.59 | 9.31 |
| RMSE | QM_MONTH | 1.15 | 2.76e-05 | 23.78 | 12.94 | 5.34 |
| RMSE | T-CLIM | 1.15 | 2.85e-05 | 24.27 | 13.45 | 5.24 |
| PBIAS [%] | CLIM | 0.6 | 2.6 | -13.9 | 5.4 | 21 |
| PBIAS [%] | QM | -0.1 | -20.1 | 9.1 | -3.3 | -2 |
| PBIAS [%] | QM_MONTH | -0.1 | -20.2 | 9.1 | -3.3 | -2 |
| PBIAS [%] | T-CLIM | -0.1 | -20.3 | 9 | -3.4 | -1.9 |





**Table 5.** Statistics from Figure 6 evaluation comparing TopoSCALE results against IMIS station data for the historical period.

| Variable | Mean IMIS | Mean TopoSCALE | R | Bias | RMSE |
|---|---|---|---|---|---|
| TA [K] | 273.8 | 273.7 | 0.998 | -0.12 | 0.63 |
| HS [m] | 0.70 | 0.63 | 0.995 | -0.07 | 0.095 |