# Peer review of "TopoCLIM: Rapid topography-based downscaling of regional climate model output in complex terrain v1.0"

_Geoscientific Model Development, 2021_

## Author Response (AR1)

**AUTHORS RESPONSE**

**REVIEWER #1: Richard Essery**

We thank the reviewer for his thoughtful comments which have undoubtedly helped to improve this work. We also apologize for the somewhat delayed response. We provide here a point-by-point response where reviewer comments are in bold type and author responses in non-bold. Changed or inserted text is given in italics.

This is a useful and well-written paper. The text is rather compressed and benefits from being read alongside Fiddes and Gruber (2012 and 2014), but there would be a lot of repetition otherwise. I have minor comments.

5 GMD guidance is that citations should not be included in the abstract unless urgently Required.

We have removed the citation.

83 It is not clear at this point what a "subgrid scheme" is.

We have clarified this as:

*"and additionally generalising this to a map product using the subgrid scheme, TopoSUB (Fiddes & Gruber 2012), which efficiently spatialises 1D model results to a map domain according to important dimensions of land surface heterogeneity."*

84 It is not clear what "repeat (ability to be experimental)" means.

Clarified by changing the line:

*"(ability to be experimental)" to*

*"(i.e. the ability to rapidly repeat numerical experiments that have a relatively low computational cost)"*

100 31 September should always be missing.

Corrected to September 30th.

136 It is not observations that are temporally downscaled.

Changed section title  2.3 to *Spatial downscaling of reanalysis data*

Changed section title 2.5 to *Temporal downscaling of climate time series*

145 FSM is a multi-physics ensemble model, but Figure 6 suggests that a single configuration is run. Which one?

We always run configuration 31. We clarify this at first introduction on l.145 as:

*"We use the Factorial Snow Model (FSM) model (Essery, 2015) to simulate the snow cover and TopoSUB (Fiddes and Gruber, 2012) to spatialise results to a 2D map. FSM is a multi-physics ensemble model, however, in this study, we always use configuration 31 which is the most complex version of the model where all five parameterisations are switched on. TopoSUB is.."*

150 It would not take much space to state the predictors in TopoSUB here.

Added as suggested:

*"Predictors used in the clustering algorithm are elevation, slope, aspect and sky view factor."*

186 ERA5 resolution is stated as 30 km here and 25 km later. It is stored on a 0.25 degree grid (and 37 vertical levels, not 137).

Corrected to 0.25 degrees at all occurrences. It's a good catch as documents state https://www.ecmwf.int/en/forecasts/datasets/reanalysis-datasets/era5 137 model levels are computed, but as you say only 37 pressure levels are stored/ made available. We clarify as:

*"vertical model levels to 137 (37 pressure levels are stored)"*

225 "residual error"

We have corrected this typo.

273 It would be better to mention that avalanching is accounted for when stating that wind transport is not in 3.4.

Edited 3.4 as suggested:

*"A higher resolution DEM may also be used but likely does not add value as processes such as wind transport that operate on these scales are not included in the model (Mott et al., 2018). However, it should be noted that avalanching off steep slopes is accounted for by removing snow linearly above a slope threshold (c.f. Fiddes et al., 2015). Importantly…"*

And edited 5.3 for consistency

*"An interesting observation in this figure is that in all time periods and scenarios snow depth is limited both at low elevation by temperature and at high elevation by terrain, which tends to be steeper and therefore permits lower accumulations due to avalanching (permitted by the model, as discussed in Section 3.4). A final point.."*

276 "upper limit of seasonal snow limit" – lose a limit

Edited to *"upper limit of seasonal snow"*

334 With downscaled reanalyses used as the reference for bias correction, how can the scheme be used to bias correct historical reanalysis?

We see this as analogous to what we do with climate data - just going backward in time with a dataset that has a special set of uncertainties due to a reduction in the number of observational datasets that are used to constrain it. We clarify this point by changing the sentence starting on Line334 to:

*"As a final point, TopoCLIM can be used to bias correct any dataset that partly overlaps with the reference period. Thus, in addition to the future projections considered in this study, it would also be possible to use TopoCLIM to correct coarser-scale reanalysis data that stretch far back in time. A prime example would be ECMWF's 20th Century reanalysis (ERA-20C) which spans 1900-2010 and thus partly overlaps with ERA5 (1950-today) that is used to drive TopoSCALE to generate the reference data T-MET."*

Figure 1

Should there also be an arrow from the downscaled reanalysis to the hourly Disaggregation?

Yes good point, the hourly TopoSCALE data is used to temporally disaggregate the quantile mapped CORDEX. We have edited the figure to reflect this and other reviewers' comments.

Figure 3 Use superscripts in units.

We have corrected the unit superscript here.

Figure 6 The time axis is not day of calendar year. Is it day of water year?

Yes this is water year and has now been clarified on axis and caption as *"Day of water year (DOWY)"*

Why is there a downward spike in snow height around day 115?

This appears to be driven by a rise in temperatures in the TopoSCALE derived ERA5 data at this point. Day 115 corresponds to December 25th in case the reviewer was concerned there was a discontinuity in the forcing related to change of year. Interestingly, the authors' combined experience suggests that Christmas day is often bad skiing conditions in the Alps.

Figure 7 There is no need to extend the scale to negative snow depths.

This is an artifact of the plot routine (R package RasterVis/Levelplot) and has been corrected. We have confirmed that there are no erroneous negative values.

Table 1 "CF long name"

We have corrected this typo.

**REVIEWER #2**

We thank the reviewer for his thoughtful comments which have undoubtedly helped to improve this work. We also apologize for the somewhat delayed response. We provide here a point-by-point response where reviewer comments are in bold type and author responses in non-bold. Changed or inserted text is given in italics.

As a general comment, this manuscript lacks a clear explanation of to what extent the method used is novel. The interest and aims of the method are clearly described, but its novelty and added-value (compared to Fiffes and Gruber, 2012, 2014, but also to previous similar downscaling and bias-adjustment methods) is not. A corollary comment is the fact that the method and results would benefit from being put into a larger context. For instance, results in section 5.3 should be compared to other studies of the impact of climate change on Alpine snow cover (which would also better fit with the current title of section 5.3 about "Alpine" snow cover, while currently only Swiss snow cover is discussed).

Useful references include (but are not limited to):

- - Steger et al. (2013): https://doi.org/10.1007/s00382-012-1545-3
- - Marty et al. (2017): https://doi.org/10.5194/tc-11-517-2017
- - Frei et al. (2018): https://doi.org/10.5194/tc-12-1-2018
- - Verfaillie et al. (2018): https://doi.org/10.5194/tc-12-1249-2018

The main novelty is that a modular system to generate high-resolution (slope scale) forcing data for impact studies is developed, documented and provided. Even though the individual components (TopoSUB, TopoSCALE, and quantile mapping) are perhaps well established, establishing the coupled workflow is novel and generates important model forcings that are currently, to our knowledge, not available to impact modellers. In particular, it allows us to get climate impact projections for the cryosphere (in this case snow) at the hillslope scale by feeding the bias corrected future atmospheric forcing through cryospheric models (in this case FSM). Perhaps most importantly, especially for applications in data scarce regions, this scheme does not require in situ data. The hillslope scale (order 100 m resolution) is often not addressed in previous climate change studies despite its importance in regulating the stores and fluxes of water, energy, and carbon (e.g.: Fan et al., 2019,https://doi.org/10.1029/2018WR023903 ), precisely because of the lack of forcing availability at this scale.

In particular we have made the following edits in the introduction to make these points clearer:

*This is especially the case in heterogeneous terrain such as mountain regions where topographic variability is high over short horizontal distances. High surface variability requires modelling at the hill slope scale (c.100 m) in order to adequately capture fluxes and stores of energy, water and carbon (Fan et al. 2019). Various methods of downscaling can be utilised to achieve this goal.*

*In this study we address the problem of impact-model ready (i.e. hillslope scale) climate timeseries with a new modelling framework called "TopoCLIM".*

*Importantly, using these pseudo-observations we are able to debias climate timeseries in regions lacking ground observations.*

Based on this and Reviewer 3 comments we have adjusted Section 5.3 title to a more appropriate: *Climate change impacts on Alpine snow cover across Switzerland*

In Section 5.3 we add discussion on previous studies while making the point that they are not comparable to our results in a straightforward manner due to resolution ( 25km v 100 m), parent climate models (ENSEMBLES v CORDEX) and/or scenarios (ie SRES v RCPs) used. However, these previous studies do nicely highlight the gap we try to fill with this work:

*Several previous studies have investigated the impacts of climate change upon Alpine snow cover (e.g. Steger et al. 2012, Marty et al. 2017, Frei et al. 2018, Verfaillie et al. 2018, Bender et al. 2020), however direct comparison is often problematic due to model resolution, analysis period, parent climate models and/or emissions scenarios used. This highlights the importance of model intercomparison studies whereby these important variables controlling model results can be standardised. Comparison to these previous works highlights the contribution of this study in that most were conducted at RCM resolution of 25 - 12 km (Steger et al. 2012, Frei et al. 2018) or local scale (Verfaillie et al. 2018, Bender et al. 2020) or reliant on in situ data (Marty et al. 2017, Bender et al. 2020). In this study, we demonstrate a method that generates results over large modelling domains at hillslope scale (100 m), a scale which is extremely important in regulating the stores and fluxes of water, energy, and carbon (Fan et al., 2019), and therefore critical to modelling snow cover in mountainous terrain. Additionally, this approach does not rely on in situ data and therefore is appropriate for data-scarce regions.*

Specific comments

Several sentences lack commas, which would facilitate the reading, e.g. line 37 before "e.g.", line 43 before "which", line 109 before "e.g.", line 127 before "it does not", line 133 before "e.g.", line 152 before "i.e.".

Corrected these and other occurrences.

Line 52: repetition of "the".

Removed.

Lines 47-58: to me, a key reference for quantile mapping that should be cited here somewhere is the pioneering work of Déqué et al. (2007):https://doi.org/10.1016/j.gloplacha.2006.11.030

Thank you for this reference which we now have included in this section.

Line 92: the "required forcing variables" are not listed here. In fact, they are listed in Table 1, but Table 1 is never called in the text…

We added the reference to the section describing the climate data, L.178-79 now reads:

*"A full description of CORDEX variables is given in Table 1 and model chains used in Table 2."*

Line 98: this sentence seems incomplete. Please edit.

Edited to:

*For example, during the conversion from a "360-day" to a "standard" calendar, the output from the linear scaling will result in a 365 day timeseries (in the case of non-leap year) and be missing the following dates: January 31st, March 31st, June 1st, July 31st, September 30th and November 30th.*

Line 104: "NWP" should be defined here.

Defined now as Numerical Weather Prediction (NWP).

Line 116: "IMIS" should be defined here instead of line 203-204.

Defined now at l. 116 as suggested. Edited both lines for consistency.

Section 2.4: Am I correct in thinking that the quantile mapping method you employ is univariate (i.e. the different variables are corrected independently from each other)? If so, this should be stated here and its implications discussed in section 5. Indeed, if temperature and precipitation are corrected independently from each other, this could have potentially large impacts on snow cover estimates (important in mountainous areas and presented in section 5.3). Could you please explain if you found a way to circumvent this issue (in section 2.4) and the related uncertainties (in section 5)?

Yes this is correct, but variables are corrected towards a physically consistent dataset in the form of downscaled ERA5 data, so we argue that while the method is univariate it will not produce physically inconsistent results. The validation during the current climate also supports this claim (Table 3-5). This is now stated explicitly in 2.4 as:

*"It should be noted that while the variables are bias corrected independently, they are corrected towards a physically consistent dataset in the form of downscaled ERA5 data, so we argue that while the method is univariate it does not produce physically inconsistent results. The validation during the current climate also supports this claim (Table 3-5)."*

Line 171 and elsewhere in the text: a distance value is always separated from its unit by a space, e.g. "44 km", "20 cm" (line 249), "100 m" (line 258), etc.

Corrected all occurrences.

Line 174: "the number of file downloads is large".

Corrected the grammar.

Section 5.1, Figure 3 and Tables 3-4: please use a consistent naming convention for your experiments. I found conflicting occurrences of "QM", "QM_QM" and "QMAP", "QM_MONTH" and "QMAP_MONTH", and "T-MET" and "T-CLIM". At line 218, you could include the acronym "CLIM": "CORDEX ensemble mean (CLIM)".

Edited for consistency.

Line 225: there is a typo ("residual error").

Corrected.

Line 227: Please remove the capital letter in "Percentage".

Corrected.

Line 244 and caption of Figure 8: "time period".

Corrected.

Line 245: "The shortest station record is 10 years, therefore the dataset nominally represents the period 1996-2018": I don't understand this statement…

Edited for clarity as:

*"IMIS station measurements are given as reference, however it should be noted that the time-period covered by each station is variable within the period 1996-2018. The shortest station record is 10 years."*

Line 275: I would split this sentence into two parts after "HIST".

Sentence split as suggested.

Line 279: please remove the capital letter in "Century".

Corrected.

Line 308: I would also split this sentence after closing the parenthesis.

Corrected.

Line 334: "historical reanalyses" (plural).

The sentence has been changed with respect to reviewer 1 comments and the plural form is no longer used.

Caption of Figure 4: "The coloured envelopes indicate".

Corrected.

Caption of Figure 5: what does "WFJ2" stand for?

We have mistakenly used the ID for the Weissfluhjoch station here - but it is confusing in this context. We have made this consistent by calling it Weissfluhjoch station as elsewhere.

Figure 6: please change the colour palette for this figure as it is not colour-blind friendly. Currently, the colours for HIST and RCP8.5 cannot be distinguished from each other by a colour-blind person. Why not use the same colours as in Figure 5?

As suggested we have adjusted the palette based on Fig 5.

Figure 7: what do "Hist_1981_2010.1" and "Hist_1981_2010.2" stand for? Isn't there only one historical scenario used in this study? If so, why are the top left and top right panels slightly different? I don't think this is explained anywhere in the text or the figure caption.

They are indeed identical datasets, however, due to the png output resolution, the output device (x11)  appeared to do some sub-pixel interpolation that gave the slight appearance of slightly different datasets. The .1 and .2 was an artefact of plotting the same dataset twice in R package RasterVis (levelplot function) - the default naming scheme appends a .1 and .2 in this case. We have replotted at a higher resolution which has removed this artefact and we have corrected the panel titles.

**REVIEWER #3**

We thank the reviewer for his thoughtful comments which have undoubtedly helped to improve this work. We also apologize for the somewhat delayed response. We provide here a point-by-point response where reviewer comments are in bold type and author responses in non-bold. Changed or inserted text is given in italics.

Code:

The provided methods could be highly useful for many people requiring downscaled climate model data for running impact models. Unfortunately, the source code in its current form is in my opinion not yet usable for the community.

First, there is no manual. The GitHub repository only contains a very short Readme explaining how to download the model and run the examples, but no information whatsoever on how to apply the methods for one's own purposes. Also, the code is quite messy. Most importantly, the parameters of the methods should not be set directly in the source files (parameters and code should always be separated (not only) for reusability and reproducibility reasons). Ideally, the code should be provided in the form of a Python package e.g. with individual functions for each method which accept their respective parameters as arguments. This would allow to run the methods from Python directly instead of having to manually execute a number of Python scripts in a particular sequence. Finally, even getting the example to run is not straightforward with the provided instructions. Running it seems to require an installation of R including the packages ncdf4 and qmap (which I found out by trial and error), but I still could not get it to run. It crashes when trying to run the R script qmap_hour_plots_daily_12.R: Error in cordex_dates_cp : object 'cordex_dates_cp' not found subprocess.CalledProcessError: Command '['Rscript', '../rsrc/qmap_hour_plots_daily_12.R', '../examples/', '_1D', '_1D.csv', '9.809', '46.83', '../examples//cordex']' returned non-zero exit status 1.

I would encourage the authors to rework the code and to provide a proper manual (or extended Readme) containing at least a description of the individual methods and how to run them, their parameters, as well as the required prerequisites apart from the Python libraries (R + packages, cdo etc.) and the supported Python versions.

We realise that the code was not only poorly documented but focused on the analysis presented in the paper. We have now done the following to make the code more clear, simple and generic to user needs, including:

- Resolved dependency issue, all dependencies (except R) are now handled by conda package manager (dependency related issue was responsible for crash the reviewer experienced)
- Reduced dependencies to make the install smoother.
- Reworked code (removed all paper specific routines (eval plots etc) which are not generic.

- Simplified ESGF helper routines so they are more generic for application to other CORDEX domains and removed inscript parameters in these helper scripts.
- Documented functions using docstrings (in script) .
- Reworked and extended the README.
- Added Python version information.

Paper:

General comments:

The paper is concise and well-written. My only major remark is that in several parts of the paper it is not immediately clear which parts are "hardcoded" in the TopoCLIM methods and which are only exemplarily used for the particular snow modeling application. E.g., does the scheme currently only allow to use CORDEX and ERA5 data or is it already possible to use other data sets (as mentioned in the conclusions)? It would be very helpful to the reader if this would be made more specific (see also my individual remarks below).

We hope we have addressed this adequately in the various specific comments below. As a very general response the basic principle of generating a pseudo-observation using TopoSCALE and using this to downscale climate data to produce forcings suitable for driving an impact model are of course (in principle) generalisable to other climate model datasets and other reanalysis. However the devil is in the details, as some of the current code base is data-management and I/O which of course is specific to the used datasets and data structures. Nonetheless, CMIP6 could for example be accessed and preprocessed without much additional effort due to a similar data structure to CORDEX. However, this step has not been yet made. We would point out that CORDEX and ERA5 permit global studies to be conducted and therefore of wide interest, as is. In the future a generalisation of the methods to accommodate other datasets (with a plugin structure) would be worthwhile.

Remembering the several schemes/data sets and their acronyms (CLIM, T-MET, T-CLIM, QM, QM_MONTH, …) can get quite challenging; maybe consider adding a table listing all of the acronyms and their meaning.

We have added the data set acronyms to Figure 1 which hopefully adds clarity and concentrates this overview in a single Figure. We hope this makes an additional table unnecessary.

Both "snow height" and "snow depth" are used throughout the paper, ideally this should be consistent.

Changed to snow depth throughout.

Specific comments:

Title: is the first dot in "v.1.0" intentional or should it be v1.0?

Edited to v1.0.

Abstract: references should be avoided in the abstract.

Removed.

Section 2.1:

When reading this, it is not really clear if the procedure refers to a single CORDEX grid point or if the method considers all grid points within a specified region.

It applies to however many grid points are in a given domain. In fact the spatial model is defined by the TopoSCALE procedure which could generate forcings for a single or multiple points on the Earth's surface. These are associated with the closest CORDEX grid centroid on a nearest neighbour basis for the quantile mapping step. This extends the same procedure described in Fiddes et al. 2015 to climate data. We have tried to clarify this in the text as:

*"and additionally generalising this to a map product using the subgrid scheme, TopoSUB (Fiddes & Gruber 2012), which efficiently spatialises 1D model results (multiple subgrid samples per CORDEX gridbox) to a map domain according to important dimensions of land surface heterogeneity. In this way, hillslope resolution (100 m) map results are generated with a laptop-feasible number of subgrid simulations (in this case 100) per large scale CORDEX gridbox. "*

Section 2.2:

It would be helpful to list exactly which preprocessing steps are performed with the CORDEX data instead of providing only some examples. This could be done in the text and/or integrated in Fig. 1 (i.e., listing all the steps instead of the generic "Preprocess CORDEX" block).

We actually don't do more than is described so this was a slightly misleading use of "such as". We have edited for clarity:

*"All preprocessing of raw CORDEX data (Figure 1): concatenating NetCDF time series, extracting region of interest and regridding from rotated pole projections, is accomplished using standard tools from the Climate Data Operators (CDO) suite."*

We haven't, however, added this to Fig 1 as we feel that this extra detail clutters what is meant to be a high-level overview - to make the conceptual flow of the scheme clear. We think that giving this detail in the text is sufficient.

"The CDO tools are incorporated into …" is a little ambiguous – I assume the tools aren ot directly integrated into the package but have to be pre-installed and are called from within the package?

Yes this is correct, we have edited the text to reflect this as:

*"The CDO tools are called from the preprocessing module of TopoCLIM and not used as standalone command line tools."*

Section 2.4:

Again, here it is not clear if the two time periods (1980-1995 and 1996-2006) are fixed in the method or if these are only used for this particular study.

These are just used in this study. Clarified as:

*"It should be noted that these periods are constraints imposed by the datasets used in this study and can be changed in other applications of the method."*

Section 2.5: Please consider adding some more detail about the used disaggregation functions for the different variables (as these can likely have considerable impact on the impact modeling results).

We have added the methods used to Table 1 as this seems to make most sense. To accommodate this extra column we have removed the CF standard name column, as this was somewhat redundant (we still give the CF long name).

"An adapted version of the Melodist package" - adapted in which way?

This was written somewhat inaccurately since we used the melodist package as is. We have edited the sentence to simply:

*The "Melodist" package is used for this purpose \citep{Forster2016-xx}.*

However not all variables are covered by Melodist (air pressure and incoming longwave) so we augmented Melodist with our own methods, as now described in the text:

*"Melodist does not provide methods for air pressure or longwave radiation, these are handled with the following procedure: (1) taking advantage of the relationship between incoming longwave radiation (ILWR) and air temperature (TA):*

*ILWR = $\epsilon$ * $\sigma$ * TA^4*

*where $\sigma$ is the Stefan-Boltzmann constant (5.67×10−8 W/m2·K4) we diagnosed the daily all sky emissivity ($\epsilon$). We then used $\epsilon$ as a daily scaling factor to convert disaggregated TA into ILWR. This procedure therefore assumes a constant $\epsilon$ at sub daily timestep (which of course will not normally be true) yet ensures that ILWR scales correctly with TA. Therefore higher TA lead to higher ILWR values and vice-versa. Air pressure is simply linearly interpolated to the sub-daily timestep."*

Section 2.6:

Again here it is not clear if this is part of the TopoCLIM method or only part of the example application for using TopoCLIM-generated data in an impact model. Since I assume the latter is the case, this section (along with 2.7) should probably be moved e.g. to section 4?

We have moved both Section 2.6 and 2.7 to Section 4 and agree it makes more sense like this to separate out core methods and study setup.

"Typical setups use …" - perhaps add some more details about a typical setup (region size, resolution).

To give a concrete example we have added: *"for example to produce the results given in Figure 7."*

Section 3.2:

What does "globally" (L172) mean in this context?

It means that while the high resolution dataset would of course be advantageous, the fact that 11 km CORDEX is not available globally (as compared to the 44 km product) means that it does not fit the design spec of what we try to do in developing and testing methods that can be used anywhere on Earth (=globally in this context).

Section 5.1:

Last paragraph: the agreement seems to be good for only some of the years in the mentioned periods. However, individual years should probably generally not be compared to observations for both the historical and the scenario period.

We have added the following text to Section 5.1:

*Available snow depth observations from WFJ show good agreement with both the historical period and first decade of the RCP runs in terms of lying within the model ensemble. It should be stressed that we do not attempt a quantitative comparison here with individual years as CORDEX variability variability(and climate models more generally) is not expected to be perfectly synchronised with observed variability at such temporal resolutions.*

Section 5.2:

If I understand correctly, DOY is used here as "days since September 1"? If so, this is really confusing, since the term DOY has a very specific meaning (with 1 being January 1). I suggest to either use the actual DOY in Fig. 6 or use another term instead of DOY.

True, we have changed both axis and caption to *day of water year (DOWY)*

Section 5.3:

The section title is confusing, since the previous results also already included TopoCLIMresults and climate change impacts on the Alpine snow cover (albeit on the point scale).

We agree this was somewhat confusing - to address this we have changed the section titles to be more precise:

*5.1 Evaluation at the Weissfluhjoch station*

*5.2 Evaluation for the IMIS network*

*5.3 Climate change impacts on Alpine snow cover across Switzerland*

"by coupling TopoCLIM with the TopoSUB spatial framework" - and with FSM in between, correct?

Yes correct, we clarified the sentence to:

*"As an example application of the full model pipeline, the results in Figure 7 were generated by feeding model results (TopoCLIM/FSM) to the TopoSUB spatial framework to generate transiently modelled snow depth maps at 100 m resolution."*

Fig. 1:

As stated above, this figure contains some very specific terms and dates (CORDEX,

ERA5, 1980-2100, 1980-2020, 90 m, 1980-2100). Since the figure should be a general overview of the TopoCLIM method I would remove all terms and dates which are not "hardcoded" in TopoCLIM.

We prefer the detail given as it gives an overview of date ranges used in the study, which we think is important for the high level overview that Figure 1 aims to achieve. However, the point is well taken and we have edited the caption to make it clear that these are indicative for this study:

*"Figure 1: Schematic overview of the main TopoCLIM processes and experimental setup used in this study."*

There is a typo in "Qantlle map".

Corrected

Fig. 2:

"with CORDEX grid overlaid" - please specify which grid (44 km?).

Edited to: *"with CORDEX EUR-44 grid (44 km) overlaid"*

Fig. 6:

The term TSCALE appears only in this figure. Probably this should be T-MET?

Corrected to T-MET in legend.

Fig. 7:

The figure has a very poor resolution.

We have improved the resolution, this also addresses plot artefact noticed by reviewer 2.

Figure caption: "Example TopoCLIM climate change maps" - maybe rephrase, since the maps are not a direct TopoCLIM result but a combination of TopoCLIM, FSM and TopoSUB.

Edited to: *"Example TopoCLIM forced climate change maps:"*

Table 1:

The table is very useful, but as far as I see it is not referenced from the paper. Consider adding a reference e.g. at the beginning of the results section.

We added the reference to the section describing the climate data, L.178-79 now reads:

 *"A full description of CORDEX variables is given in Table 1 and model chains used in Table 2."*

What do the timesteps of 3 and 6 hours mean?

This is a legacy mistake from previous work. Timestep has been corrected to "daily" in all cases.

Minor remarks:

P3 L65: Add reference to ERA5 here (currently in section 3.3) if this is the first Mention.

Moved reference as suggested.

P3 L69: Either write "a subgrid clustering scheme" or "the subgrid clustering scheme TopoSUB".

Corrected to the second suggestion

P4 L118: Add a colon at the end of the line.

Done.

P5 L130 and P8 L228: replace QMAP_MONTH by QM_MONTH as in the rest of the paper (assuming this does not actually refer to a different variant).

Corrected for consistency (as addressed also by Reviewer 2).

P5 L145: Remove the second "model".

Done.

P6 L173: Capitalize "python".

Done.

P7 L202: Adding "(cm)" is probably not necessary.

Removed.

P7 L207: Replace 2031-60 and 2070-100 with 2031-2060 and 2070-2100.

Done.

P8 L219: The term "QM_QM" appears only here and in the caption of Fig. 3, I assume this is what is referred to as only "QM" in the rest of the paper?

Corrected for consistency (as addressed also by Reviewer 2)

P8 L225: Typo ("erro").

Corrected.

P10 L283: "25 km" -> not 30 km (as stated in section 3.3)?

Corrected to 0.25 degrees throughout.

P10 L293: The acronyms (TA, ILWR, ISWR) appear here for the first and only time; please write the full variable names.

---

## Author Response (AR2)

**Authors response**

**Reviewer 1**

**Manuscript:**

**Section 3.2: regarding my earlier comment on "globally": First, perhaps consider rewording "this is not available globally" by "this is not available for all CORDEX regions" or something similar. Second, it is still not clear to me how using a higher resolution product would not be "fit for the purpose of this study" (as stated in the manuscript), if it is a regional study for Switzerland? Is the 44 km resolution currently hardcoded into topoCLIM?**

The method presented aims to be a global method based on datasets only available worldwide - we test the approach in Switzerland but envisage applications in much more data poor regions, hence why we use the 44 km over the only regionally available 11 km. The 44 km product is not hardcoded, the code can accept any CORDEX resolution.

We have changed the sentence as suggested and added some clarification for the second part of the comment:

*"this is not available for all CORDEX regions, and therefore not fit for the purpose of this study which aims to be a globally applicable method based on datasets that are available worldwide."*

**L27: broken reference ("alias?")**
Thanks - fixed.

**L94: Replace colon after "All preprocessing of raw CORDEX data" by a hyphen (-> "All preprocessing of raw CORDEX data - … - is accomplished …")**
done

**L190: remove opening quotation mark**
done

**L192: remove double parentheses**
done

**Code:**

**The instructions for installing the package and running the example setup are now comprehensive and clear to follow. Only running the command Rscript -e 'install.packages("ncdf4","qmap," repos="http://cran.us.r-project.org")' results in an error; I had to change it to Rscript -e 'install.packages(c("ncdf4","qmap"), repos="http://cran.us.r-project.org")'. Furthermore the documentation contains some typos (e.g. "datsets", "writted").**

- We have corrected the packages install line.

- We have corrected these typos and read through both readme documentations and script docstrings again.

**The code cleanup and the introduction of comments and docstrings has improved the code a lot. Some parameters (e.g. (1) the lat/lon ranges in esgf_get.py or the (2) dates in qmap_hour_plots_daily_12.R) are however still hardcoded - ideally these should also be passed by the user, but at the minimimum you should add a remark in the documentation where/how to change them.**

(1) These are actually indexes so have refactored the variables and added as arguments to esgf_get.py as follows:

*ARGS:*
*cordex_domain (str): cordex experiment domain e.g. "EUR-44"*
*openid (str): Your openID as configured at ESGF e.g. 'https://esgf-data.dkrz.de/esgf-idp/openid/xxxx'*
*outdir (str): Path to write results to e.g. /path/to/results*
*xstartIndex (interger): Start index slice in x direction (left to right) to reduce array size for download*
*xendIndex (interger): End index slice in x direction (left to right) to reduce array size for download*
*ystartIndex (interger): Start index slice in y direction (top to bottom) to reduce array size for download*
*yendIndex (interger): End index slice in y direction (top to bottom) to reduce array size for download*

(2) We have introduced a config file in "ini" format to hold the various date parameters. This promotes reproducibility as this file can be stored alongside results to reproduce results easily or be committed to a version control repository. This also permits additional parameters that could be added in future releases to be easily included. We have added a section in the documentation to describe how this works and defined the parameters.

Work on code for this revision corresponds to the following commits (24 Jan 2022):

cfaa7d5bb9cd9994ea661793407ff76c91524f13
bf43a4c35e595c13d9c449b2b3b277022a82dbf7
933221caf4af463dd5fa7c92bd68c3f121fad957

**Editors comments**

**I, for my part, will ask you to make sure that your code & data availability section is up-to-date, in particular I think that the code DOI (and title!) may need an update to version 1.1 or similar.**

We have created a new code repository with a new DOI on the WSL institutional data repository Envidat at https://www.envidat.ch/dataset/topoclim-v1-1-code. This is linked to v1.0 repository. The code availability section is updated with this new DOI. The manuscript title has also been updated to v1.1.